# Fatal heart disease among cancer patients

Kelsey C. Stoltzfus [1], Ying Zhang[2], Kathleen Sturgeon[3], Lawrence I. Sinoway[4], Daniel M. Trifiletti[5], Vernon M. Chinchilli[3] & Nicholas G. Zaorsky [1,3 ✉]

As the overlap between heart disease and cancer patients increases as cancer-specific mortality is decreasing and the surviving population is aging, it is necessary to identify cancer patients who are at an increased risk of death from heart disease. The purpose of this study is to identify cancer patients at highest risk of fatal heart disease compared to the general population and other cancer patients at risk of death during the study time period. Here we report that 394,849 of the 7,529,481 cancer patients studied died of heart disease. The heart disease-specific mortality rate is 10.61/10,000-person years, and the standardized mortality ratio (SMR) of fatal heart disease is 2.24 (95% CI: 2.23–2.25). Compared to other cancer patients, patients who are older, male, African American, and unmarried are at a greatest risk of fatal heart disease. For almost all cancer survivors, the risk of fatal heart disease increases with time.

---

[1] Department of Radiation Oncology, Penn State Cancer Institute, Hershey, PA 17033, USA. [2] Biostatistics and Research Decision Sciences, Merck & Co, North Wales, PA 19454, USA. [3] Department of Public Health Sciences, Penn State College of Medicine, Hershey, PA 17033, USA. [4] Department of Medicine, Penn State College of Medicine, Hershey, PA 17033, USA. [5] Department of Radiation Oncology, Mayo Clinic, Jacksonville, FL 32224, USA. ✉email: nicholaszaorsky@gmail.com

For decades, heart disease has been the leading cause of death in Americans, responsible for about one in every four deaths[1]. Deaths from cancer are starting to surpass the deaths due to heart disease[2,3]. Additionally, the overlap between heart disease patients and cancer patients is increasing as cancer-specific mortality is decreasing and the surviving population is aging[4]. This has led to the development of the field of cardio-oncology, which refers to the treatment of cardiovascular disease in cancer patients[5,6], with particular focus on the adverse effects of cancer therapy[7]. Cardiotoxicity in cancer patients became more prevalent as drugs such as anthracyclines and targeted kinase inhibitors were linked to unexpected cardiovascular outcomes, such as heart failure[8,9]. The ultimate goal in cardio-oncology is to anticipate issues related to cardiotoxicity when treating cancer patients[7]. As cardio-oncology evolves, it will become crucial to identify cancer patients who are at an increased risk of death from heart disease with the goal of developing targeted prevention strategies.

The American Heart Association (AHA) has limited recommendations regarding the prevention of heart disease in cancer patients[10]. The National Comprehensive Cancer Network (NCCN) provides guidelines addressing the stages, treatment, and surveillance of cardiomyopathy in cancer patients, in addition to recognizing the new field of cardio-oncology[11]. While small advancements have been made to address the overlapping fields of heart disease and cancer care, there is currently no comprehensive resource to assist clinicians, including primary care physicians, oncologists, and cardiologists, in identifying cancer patients at highest risk of fatal heart disease. International organizations, such as the European Society for Cardiology, recognize the need for such guidelines due to the complex nature of the relationship between cancer and cardiovascular health, and have published materials to assist clinicians in the care of this patient population until formal guidelines are available[12].

The purpose of the current work is to present an analysis of the risk of death from heart disease among all cancer patients. We define fatal heart disease as an overarching umbrella term to include a variety of diseases of the heart that led to the patients' demise. Our objectives are to identify cancer patients at highest risk of fatal heart disease compared to (1) the general population, using standardized mortality ratios (SMRs), and (2) other cancer patients at risk of death during the study time period, using odds ratios (ORs) and hazard ratios (HRs). Our overarching goal is to identify subgroups of cancer patients at greatest risk of fatal heart disease.

Here we report that 394,849 out of 7,529,481 cancer patients studied died of heart disease. The heart disease-specific mortality rate is 10.61/10,000-person years, and standardized mortality ratio (SMR) of fatal heart disease is 2.24 (95% CI: 2.23–2.25). Compared to other cancer patients, patients who are older, male, African American, and unmarried are at a greatest risk of fatal heart disease. If <40 years of age, the plurality of heart disease deaths occurs in patients treated for breast cancer and lymphomas; if ≥40, from cancers of the prostate, colorectum, breast, and lung. For almost all cancer survivors, the risk of fatal heart disease increases with time. These findings should be used to develop comprehensive guidelines regarding the prevention and care of heart disease in cancer survivors.

## Results

**Fatal heart disease, cancer patients vs general population.** A total of 7,529,481 cancer patients were included in the analysis; of these, 394,849 (5.24%) died from heart disease. Among all cancer patients, the heart disease-specific mortality rate per 10,000-person years was 10.61, and the SMR of fatal heart disease was 2.24 (95% CI: 2.23, 2.25, relative risk—$P < 0.0001$).

Table 1 shows the characteristics of all cancer patients included, as well as those who died of heart disease. Males were more likely to die from heart disease compared to females, 59.5% vs 40.5%, respectively. Patients who were diagnosed at a younger age had a higher SMR for fatal heart disease, and the SMRs gradually declined as patients were diagnosed at a later age: e.g. those <39-years had an SMR of 43.8 (95% CI: 41.0, 46.7, relative risk—$P < 0.05$) vs >80-year-olds had an SMR of 1.73 (95% CI: 1.71, 1.74, relative risk—$P < 0.0001$). Although there were only 45,412 (11.5%) patients with metastatic/distant disease at diagnosis, these patients had the highest SMR of death from fatal heart disease, 5.10 (95% CI: 5.02, 5.18, relative risk—$P < 0.0001$). Those diagnosed 1992–2000 had an SMR of 1.85 (95% CI: 1.84, 1.86, relative risk—$P < 0.0001$), and those diagnosed 2011–2015 had an SMR of 12.0 (95% CI: 11.7, 12.2, relative risk—$P < 0.0001$).

Figure 1 shows SMRs of fatal heart disease among cancer patients by cancer subsite. Overall, the risk of fatal heart disease for cancer patients is more than twice that of the general population (for all sites, the SMR at 1–5 years after diagnosis is 1.93, 95% CI: 1.91, 1.95; relative risk—$P < 0.0001$), and this risk increases with follow-up time (for all sites, the SMR at >10 years after diagnosis is 2.73, 95% CI: 2.7, 2.75; relative risk—$P < 0.0001$). Certain cancer patients have relatively high SMRs from fatal heart disease in the first year after diagnosis, including lung, myeloma, oral cavity and pharynx, kidney, and leukemia. For example, patients with lung cancer and myeloma have SMRs of 7–14 in the first year after diagnosis (relative risk—$P < 0.0001$). The SMRs for lung cancer and myeloma remain elevated throughout all follow-up times, with SMRs of 4–5 after 10+ years after diagnosis. In each of the cancer sites presented in Fig. 1, risk of fatal heart disease is greater than the general population across all time points in the follow-up period. For reference, the SMR for all cancer subsites available through SEER is available in Source Data file.

**Fatal heart disease across cancer subgroups.** Table 2 (left panel) shows the ORs of patients who died of heart disease, stratified by subgroup. Patients older than 80 years of age have a fatal heart disease OR of 44.78 (95% CI: 42.61, 47.05, chi square—$P < 0.0001$) compared to those <39 years of age. African Americans have an OR of 1.29 (95% CI: 1.27, 1.3, chi square—$P < 0.0001$) compared to whites. Patients with localized disease had a higher OR of fatal heart disease compared to those with distant metastases, OR of 2.48 (95% CI: 2.45, 2.51, chi square—$P < 0.0001$). Additional findings include a higher OR for males vs. females (OR = 1.56; 95% CI: 1.54, 1.57, chi square—$P < 0.0001$) and unmarried vs. married (OR = 1.27; 95% CI: 1.27, 1.28, chi square—$P < 0.0001$). When compared to patients diagnosed from 1992–2000, patients diagnosed in more recent years had a lower OR (2001–20015: OR = 0.62; 95% CI: 0.61, 0.62, chi square—$P < 0.0001$; 2006–2010: OR = 0.36; 95% CI: 0.36, 0.36, chi square—$P < 0.0001$; 2011–2015: OR = 0.15; 95% CI: 0.14, 0.15, chi square—$P < 0.0001$).

Figure 2 shows the heart disease-specific mortality rate of cancer patients as a function of age group. Table 2 (Cox proportional hazards model in right panel) shows the HRs of patients who died of heart disease, stratified by subgroup, complementing the results of Fig. 2. Relatively few patients <40 years of age died of heart disease, in part because most cancers are diagnosed in the elderly. Among patients diagnosed at age <40, the plurality of fatal heart disease occurs in patients with breast cancer and lymphomas. In contrast, among patients diagnosed at age ≥40, the plurality of fatal heart disease occurs in patients diagnosed with cancers of the prostate, colorectum, breast, and lung. The relative risk of fatal heart disease is dramatically higher in the

**Table 1 Standardized mortality ratios of heart disease death among cancer patients.**

| | Total[a] | Heart disease[a] | Heart disease per 10,000 Person-years[b] | SMR (95% CI)[b] |
|---|---|---|---|---|
| **Age group** | | | | |
| ≤39 | 450,691 (6.0%) | 1593 (0.4%) | 0.47 | 43.8 (41.0–46.7) |
| 40–49 | 669,634 (8.9%) | 6238 (1.6%) | 1.36 | 20.38 (19.68–21.09) |
| 50–59 | 1377,815 (18.3%) | 24,665 (6.2%) | 3.03 | 10.10 (9.93–10.28) |
| 60–69 | 1,928,342 (25.6%) | 72,539 (18.4%) | 7.18 | 3.73 (3.70–3.77) |
| 70–79 | 1864,031 (24.8%) | 143,320 (36.3%) | 17.90 | 1.93 (1.92–1.95) |
| 80+ | 1238,968 (16.5%) | 146,494 (37.1%) | 48.43 | 1.73 (1.71–1.74) |
| **Sex** | | | | |
| Female | 3,661,011 (48.6%) | 160,101 (40.5%) | 8.56 | 2.32 (2.31–2.34) |
| Male | 3,868,470 (51.4%) | 234,748 (59.5%) | 12.69 | 2.19 (2.18–2.21) |
| **Race** | | | | |
| White | 6,186,237 (82.2%) | 334,507 (84.7%) | 10.77 | 2.11 (2.10–2.12) |
| African American | 770,801 (10.2%) | 41,851 (10.6%) | 12.45 | 3.04 (3.00–3.08) |
| Other | 499,751 (6.6%) | 17,658 (4.5%) | 7.25 | 3.56 (3.50–3.63) |
| Unknown | 72,692 (1.0%) | 833 (0.2%) | 2.34 | 0.00 (0.00, 0.00) |
| **Marital status** | | | | |
| Married | 4,092,712 (54.4%) | 192,652 (48.8%) | 8.57 | 2.10 (2.08–2.11) |
| Unmarried | 2,895,946 (38.5%) | 173,304 (43.9%) | 14.35 | 2.47 (2.45–2.49) |
| Unknown | 540,823 (7.2%) | 28,893 (7.3%) | 10.88 | 2.11 (2.08–2.15) |
| **Stage** | | | | |
| Distant | 1,526,068 (20.3%) | 45,412 (11.5%) | 15.90 | 5.10 (5.02, 5.18) |
| Regional | 2,309,633 (30.7%) | 126,342 (32.0%) | 9.53 | 2.27 (2.25, 2.29) |
| Localized | 2,508,615 (33.3%) | 149,675 (37.9%) | 9.09 | 2.06 (2.04, 2.07) |
| Unstaged/unknown | 1,185,165 (15.7%) | 73,420 (18.6%) | 15.83 | 2.02 (2.00, 2.04) |
| **Year of diagnosis** | | | | |
| 1992–2000 | 1,624,977 (21.6%) | 163,648 (41.4%) | 12.75 | 1.85 (1.84–1.86) |
| 2001–2005 | 1,824,980 (24.2%) | 119,273 (30.2%) | 10.12 | 2.62 (2.59–2.64) |
| 2006–2010 | 1,992,271 (26.5%) | 77,987 (19.8%) | 8.66 | 4.38 (4.33–4.44) |
| 2011–2015 | 2,087,253 (27.7%) | 33,941 (8.6%) | 9.45 | 12.0 (11.7–12.2) |
| **Surgery** | | | | |
| Yes | 4,315,322 (57.3%) | 228,306 (57.8%) | 8.29 | 2.09 (2.07, 2.10) |
| No | 3,062,213 (40.7%) | 157,343 (39.8%) | 16.65 | 2.54 (2.52, 2.56) |
| Unknown | 151,946 (2.0%) | 9200 (2.3%) | 40.80 | 3.04 (2.84, 3.26) |
| All patients | 7,529,481 | 394,849 (5.24%) | 10.61 | 2.24 (2.23–2.25) |

[a]Data base "SEER 18 Regs Research Data + Hurricane Katrina Impacted Louisiana Cases, Nov 2017 Sub (1973-2015 varying)" was used.
[b]Data base "Incidence - SEER 13 Regs excluding AK Research Data, Nov 2017 Sub (1992-2015) for SMRs" was used; exact method was used to calculate 95% CI.

elderly: HR 80+ year-olds vs those ≤39-year-olds is 131.94 (95% CI 125.94, 138.74, $P < 0.0001$). The relative risk of fatal heart disease is also significantly higher in African Americans vs. whites (HR = 1.34, 95% CI 1.33, 1.36, chi square—$P < 0.0001$), unmarried vs. married (HR = 1.52, 95% CI 1.51, 1.53, chi square—$P < 0.0001$), and patients who did not receive surgery vs. those who did receive surgery (HR = 1.30, 95% CI 1.29, 1.31, chi square—$P < 0.0001$). Compared to patients with distant disease staging, regional and localized staged patients had a lower relative risk of fatal heart disease (HR = 0.65, 95% CI 0.65, 0.66; HR = 0.78, 95% CI 0.77, 0.79, respectively). Males had a slightly higher risk of fatal heart disease compared to women (HR = 1.48, 95% CI 1.47, 1.5, chi square—$P = 0.024$).

Figure 3 shows the relative percentage of patients who die from heart disease or the primary cancer. Each cancer site is represented in a separate graph, as indicated by the colors. For each of the cancer sites presented, the relative percentage of fatal heart disease cases increases as follow-up time increases. At 10+ years of follow-up time, more patients die from heart disease than from their primary cancer in the following sites: prostate, colon and rectum, bladder, melanoma, kidney, endometrial, oral cavity and pharynx. In contrast, patients diagnosed with myeloma are more likely to die from their primary cancer than heart disease over the entire follow-up period (10+ years: fatal heart disease = 18.2%, myeloma = 81.76%). Trend tests for changes in proportion of deaths from primary cancer versus heart disease

are statistically significant for each site tested (chi-squared value range: 30.9–123,840; $P < 0.001$), as presented in Supplementary Table 2. Additionally, Supplementary Figs. 1 and 2 display the observed number of deaths for each of the primary cancer sites presented, comparing the following cause of death groupings: primary cancer, diseases of the heart, non-index cancer, and other causes of death.

**Discussion**
We present an analysis of the risk of fatal heart disease among more than 7.5 million cancer patients. Fatal heart disease varies as a function of primary cancer disease type, follow-up time, and other clinical covariates, such as age and race. The risk of fatal heart disease is more than double in cancer patients compared to the general population. After a decline in risk within one year of cancer diagnosis, the risk of death from heart disease continues to rise as time after diagnosis increases. Odds of death from heart disease differs by age at diagnosis, sex, race, marital status, stage of primary cancer, year of diagnosis, and whether the patient received surgery. Among those diagnosed at <40 years of age, the plurality of heart disease deaths occurs in patients treated for breast cancer and lymphomas; if ≥40, from cancers of the prostate, colorectum, breast, and lung. This is the first study to assess the risk of death due to all diseases of the heart across, independent of cancer site and treatment type.

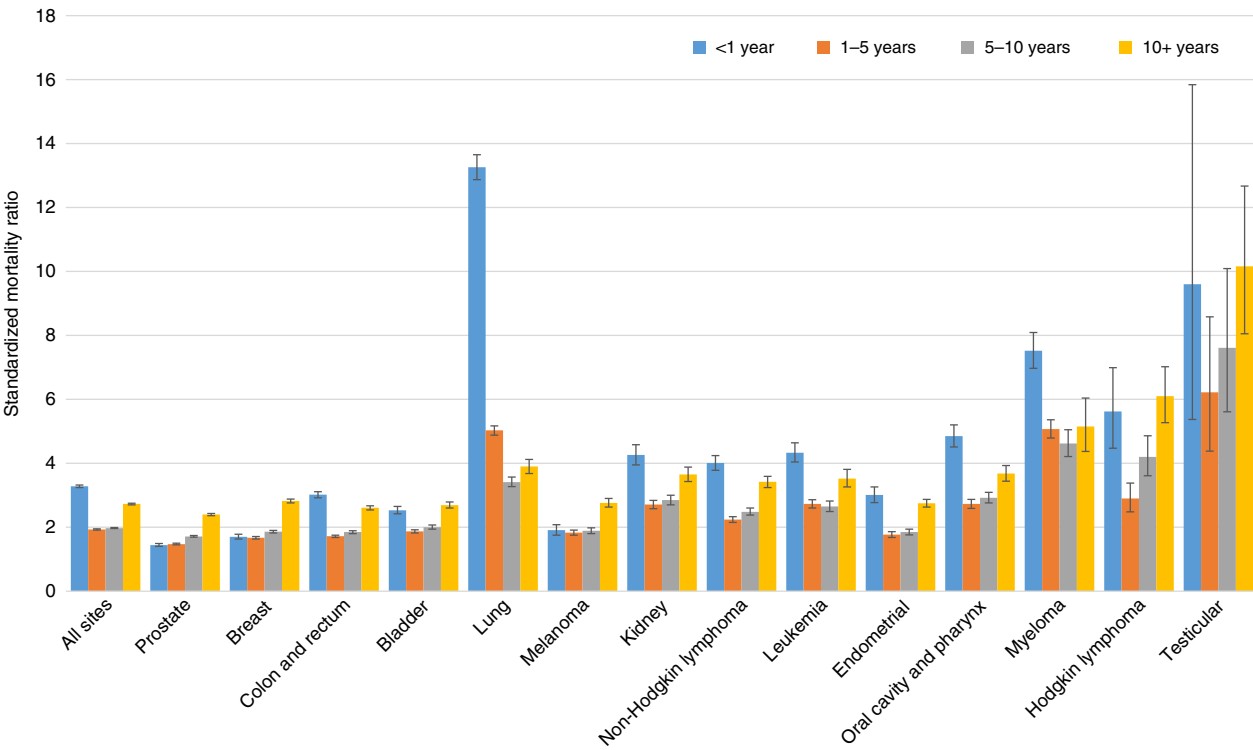

**Fig. 1 Standardized mortality ratios (SMRs) of fatal heart disease among cancer patients by cancer subsite.** The y-axis depicts the SMR with 95% CI, and the x-axis depicts the disease site. Different time periods after diagnosis (<1 year vs 1–5 years vs 5–10 years vs >10 years) are shown in blue, orange, grey, and yellow, respectively. At the 1–5 year post-diagnosis time point, the risk of heart disease among cancer patients is two times that of the general population and rises with longer follow-up time. Certain cancer patients have relatively high SMR from heart disease in the first year after diagnosis (e.g. lung with SMR of 13.3, and myeloma with SMR of 7.5). These 12 cancer were chosen as they represent the sites with the greatest number of person–years at risk. Total person years at risk = 1,195,381.08. Error bars represent the 95% CI by site. Source data are provided as a Source Data file.

Regarding objective 1, we found heart disease contributed to 5.24% of deaths among all cancer patients, and the risk of death from heart disease is 2.24 times that of the general population (Table 1, Fig. 1). Patients diagnosed with lung cancer were at a significantly increased risk of fatal heart disease within one year after diagnosis, with decreasing, yet still elevated SMRs observed throughout the entire follow-up time period. Patients with myeloma and cancers of the oral cavity and pharynx also had an elevated risk of fatal heart disease compared to the general population. This increase in risk was greatest in the first year of follow-up and remained greater than the general population over the 10 years of follow-up. For all 12 cancer sites presented in Fig. 1, risk of fatal heart disease was higher than the general population in each of the follow-up time periods, although not as pronounced in patients with cancer of the prostate, breast, bladder, and melanoma. Regarding objective 2, we found the plurality of heart disease deaths occur in patients treated for breast cancer and lymphomas in children, adolescents, and adults <40 years old. In adults ≥40 years old, the plurality of heart disease deaths occur in patients with the prostate, colorectum, breast, and lung cancer. Patients diagnosed with cancer in more recent years (2001–2005, 2006–2010, 2011–2015) had a lower odds of death from heart disease than patients diagnosed from 1992 to 2000 (Table 2). Patients diagnosed in the earliest time period have had more follow-up years, and thus more years surviving their cancer; surviving their cancer makes them more likely to die from another cause, such as heart disease. Patients with more aggressive forms of cancer diagnosed in the more recent time period are more likely to die from their cancer, and are thus no longer at risk of dying from heart disease.

Additionally regarding risk of fatal heart disease (Table 2), we observed differences between ORs and HRs specifically when looking at fatal heart disease by stage and surgery. This may be explained in part because patients with distant disease have a more aggressive form of cancer, often do not receive surgery as primary treatment, and are thus more likely to die from their primary cancer than from heart disease. Survival time is ultimately shorter in this sicker population.

When analyzing SMRs by cancer site, we observed a U-shaped phenomenon among each of the sites. There is a large SMR for the first year after diagnosis, followed by a drop in SMRs for the 1–5 year post-diagnosis category, and a rise in SMRs at 5–10 years and 10+ years. This phenomenon may be due to acute and long-term toxicities of therapy. Acutely, certain cancer therapies cause heart damage, including chemoradiotherapy delivered to the thorax for lung cancer. The acute effects of therapies may subside, but patients may also be at risk for long-term cardiomyopathy and other unexpected cardiovascular events[8,9]. For example, patients with Hodgkin lymphoma are at risk of cardiac events decades after therapy, hypothesized to be due to radiation to the mediastinum and receipt of doxorubicin[13]. Similar increased risk of heart disease post-treatment has been observed in women receiving certain forms of radiation therapy for breast cancer[14] and childhood cancer survivors who received chemotherapy or radiation therapy[15]. In aggressive forms of prostate cancer, patients may receive androgen deprivation therapy (ADT) as a component of the treatment plan[16]. While beneficial for cancer outcomes, the adverse effects of ADT on overall health include increased insulin resistance and weight gain, which are both comorbidities with heart disease[16]. Increased risk of heart failure after use of ADT is likely contributing to the increase in

**Table 2 Odds ratios and hazard ratios of fatal heart disease among cancer patients.**

| | Logistic regression model | | | Cox proportional hazards model | | |
|---|---|---|---|---|---|---|
| | Odds ratio | 95% CI | P-value[a] | Hazard ratio | 95% CI | P-value[a] |
| Age group | | | <0.0001 | | | <0.0001 |
| ≤39 | 1.00 | – | | 1.00 | – | |
| 40–49 | 2.83 | (2.68, 2.99) | | 3.65 | (3.45, 3.86) | |
| 50–59 | 5.86 | (5.57, 6.17) | | 8.25 | (7.83, 8.68) | |
| 60–69 | 12.52 | (11.91, 13.16) | | 19.13 | (18.19, 20.12) | |
| 70–79 | 25.27 | (24.05, 26.56) | | 47.73 | (45.39, 50.18) | |
| 80+ | 44.78 | (42.61, 47.05) | | 131.94 | (125.48, 138.74) | |
| Sex | | | <0.0001 | | | 0.0240 |
| Female | 1.00 | – | | 1.00 | | |
| Male | 1.56 | (1.54, 1.57) | | 1.48 | (1.47, 1.5) | |
| Race | | | <0.0001 | | | <0.0001 |
| White | 1.00 | – | | 1.00 | | |
| African American | 1.29 | (1.27, 1.3) | | 1.34 | (1.33, 1.36) | |
| Other | 0.76 | (0.75, 0.77) | | 0.77 | (0.76, 0.78) | |
| Unknown | 0.32 | (0.3, 0.34) | | 0.28 | (0.27, 0.31) | |
| Marital status | | | <0.0001 | | | <0.0001 |
| Married | 1.00 | – | | 1.00 | | |
| Unmarried | 1.27 | (1.27, 1.28) | | 1.52 | (1.51, 1.53) | |
| Unknown | 1.26 | (1.25, 1.28) | | 1.10 | (1.09, 1.11) | |
| Stage | | | <0.0001 | | | <0.0001 |
| Distant | 1.00 | – | | 1.00 | | |
| Regional | 1.94 | (1.91, 1.96) | | 0.65 | (0.65, 0.66) | |
| Localized | 2.48 | (2.45, 2.51) | | 0.78 | (0.77, 0.79) | |
| Unstaged/unknown | 1.84 | (1.82, 1.86) | | 0.88 | (0.87, 0.89) | |
| Year of diagnosis | | | <0.0001 | | | <0.0001 |
| 1992–2000 | 1.00 | – | | 1.00 | – | |
| 2001–2005 | 0.62 | (0.61, 0.62) | | 0.83 | (0.82, 0.83) | |
| 2006–2010 | 0.36 | (0.36, 0.36) | | 0.70 | (0.69, 0.71) | |
| 2011–2015 | 0.15 | (0.14, 0.15) | | 0.63 | (0.62, 0.63) | |
| Surgery | | | <0.0001 | | | <0.0001 |
| Yes | 1.00 | – | | 1.00 | – | |
| No | 0.89 | (0.88, 0.9) | | 1.30 | (1.29, 1.31) | |
| Unknown | 0.81 | (0.79, 0.83) | | 1.25 | (1.21, 1.3) | |

[a]Two-sided test; Chi square.

SMRs long-term in this population, and should thus be greatly considered in the survivor care plan of the patient[17].

The AHA provides guidelines for the primary prevention of cardiovascular disease for the general population and for adults with specific comorbidities, including type 2 diabetes mellitus, hypertension, and high cholesterol[10]. Cancer patients, are not specifically discussed regarding prevention of heart disease. However, the AHA[4] does recognize the increasing population of cancer survivors who are in need of multimodal cardiac rehabilitation programs. Further, the AHA has recognized the shared risk factors between breast cancer patients and heart disease patients, also acknowledging the cardio-toxic effects of breast cancer therapy and the need for heart disease prevention in this population[18]. The NCCN[11] provides survivorship guidelines after therapy for cancer, with the goal of preventing long-term morbidity and mortality. These guidelines address the stages, treatment, and surveillance of cardiomyopathy in addition to recognizing the new field of cardio-oncology. Despite some advancements, a comprehensive statement regarding heart disease prevention or care post-treatment has not been made for all cancer patients. International organizations, such as the European Society for Cardiology, have made strides towards the development of such guidelines and recognize the complex nature of the relationship between cancer and cardiovascular health[12]. A 2016 article by Zamorano et al. highlights the cardiovascular risks of cancer treatments, strategies for preventing cardiovascular-related complications, and the need for long-term survivorship programs

for cancer patients[12]. The current study, in combination with international guidelines, may be used to create future guidelines for the US cancer population.

As cancer treatments improve and patients are living longer, patients are more likely to die from a non-cancer cause[19]. Coinciding with our results, a 2012 study by Ward et al. found women with endometrial cancer are more likely to die from cardiovascular disease than their primary cancer, and that risk of cardiac-related death is greater than risk of cancer death at 5 years post-diagnosis[20]. The current results suggest that heart disease prevention strategies and programs should be a focus for cancer patients, specifically adults over age 40 with prostate, colorectum, breast, and lung cancer. Additional populations to target, based on the observed increased ORs, include men, African Americans, and unmarried patients. Cancer patients with less aggressive, localized disease are also at an increased risk of fatal heart disease and should thus be a focus for prevention efforts. Guidelines regarding the prevention of heart disease should be standard information provided by clinicians to patients, and be incorporated into the long-term care plan of the patient. As the field of cardio-oncology continues to develop, we recommend that organizations such as the AHA provide strong guidelines for the care of cancer patients. Specifically, our findings may assist clinicians in knowing which subgroups of cancer patients are at the greatest risk of fatal heart disease, and therefore whom should be their primary concern regarding heart disease prevention.

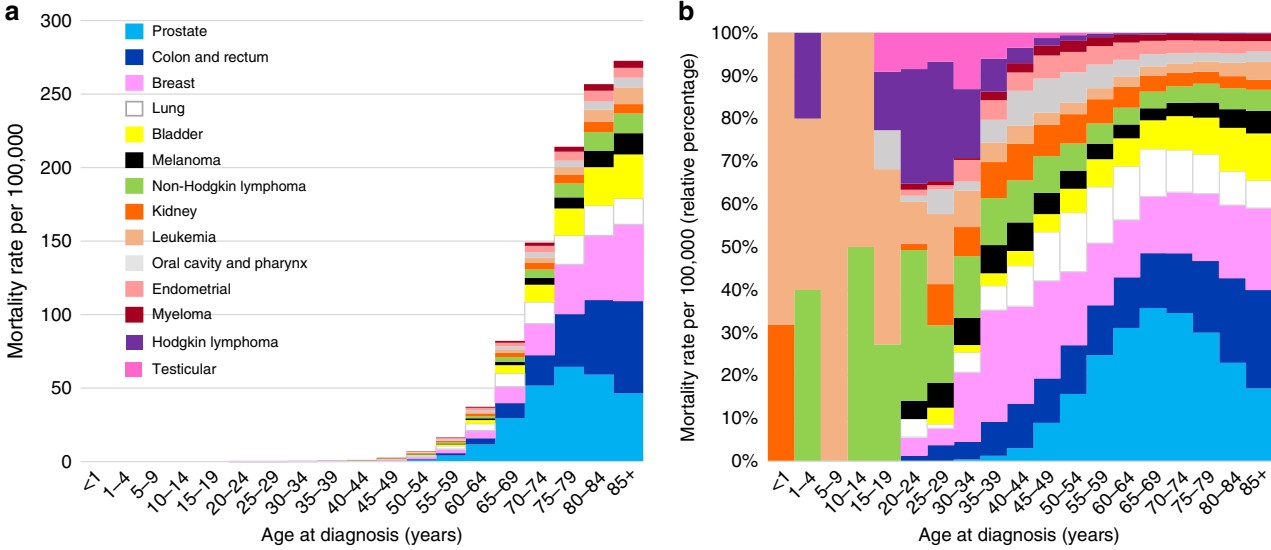

**Fig. 2 Age adjusted mortality rates per 100,000 for fatal heart disease by cancer subsite. a** The *y*-axis depicts the heart disease-specific mortality rate per 100,000 and the x-axis depicts the age group at diagnosis in years. The colors depict the disease sites as follows: light blue = prostate; dark blue = colon and rectum; pink = breast; white = lung; yellow = bladder; black = melanoma; green = non-Hodgkin lymphoma; orange = kidney; light orange = leukemia; gray = oral cavity and pharynx; coral = endometrial; maroon = myeloma; purple = Hodgkin lymphoma; dark pink = testicular. The plurality of fatal heart disease occurs in patients diagnosed with cancer of the prostate, colon and rectum, breast, and lung, and the majority fatal heart disease cases are diagnosed at an older age. **b** The *y*-axis depicts the relative heart disease-specific mortality rate per 100,000 compared to all cancer patients, and the x-axis depicts the age group at time of diagnosis. For each age group, heart disease-specific mortality rates are displayed as a relative percentage to other cancer sites in that specific age group. The colors depict the disease sites. For patients under age 40, the plurality of heart disease deaths occurs in patients treated for breast cancer and lymphomas. In contrast, among patients ≥40 years old, the plurality of heart disease deaths occurs in patients treated for cancers of the prostate, breast, and colon and rectum. Source data are provided as a Source Data file.

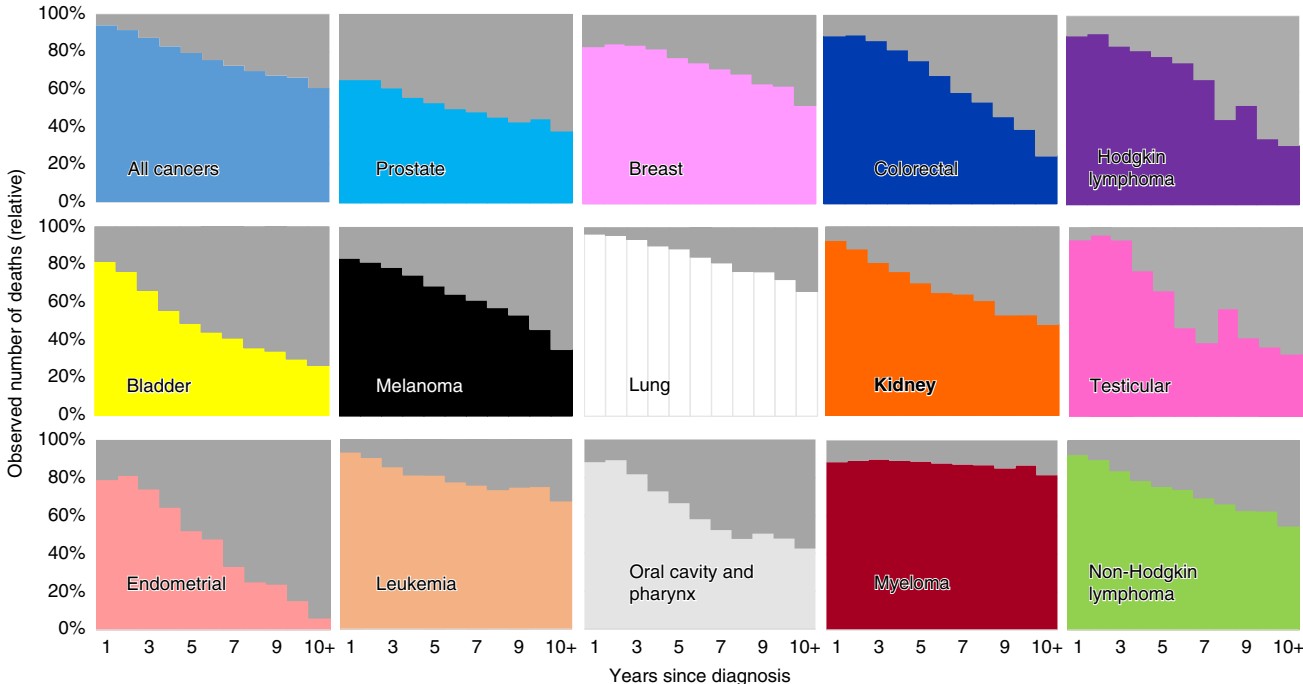

**Fig. 3 Observed number of deaths due to primary cancer vs. fatal heart disease.** The *y*-axis depicts the relative percent of cases from each cause of death. The x-axis depicts the number of years since cancer diagnosis. Each graph represents a patient of the same primary disease site, as indicted by the colors. The colors depict the disease sites as follows: light blue = prostate; dark blue = colon and rectum; pink = breast; white = lung; yellow = bladder; black = melanoma; green = non-Hodgkin lymphoma; orange = kidney; light orange = leukemia; gray = oral cavity and pharynx; coral = endometrial; maroon = myeloma; purple = Hodgkin lymphoma; dark pink = testicular. Grey represents fatal heart disease while all other colors represent death from primary cancer diagnosis. For all sites, the percentage of fatal heart disease cases increases with follow-up time. The largest relative increase in fatal heart disease cases over the follow-up period is seen in endometrial cancer. Myeloma patients are more likely to die from their primary cancer than from heart disease at all time points during the follow-up period. All trend tests for changes in proportion of death from primary cancer versus heart disease are statistically significant (two-sided test; chi-squared value range: 30.9–123,840; *P* < 0.001). Source data are provided as a Source Data file.

Our study has limitations. First, the primary variable of interest, diseases of the heart, encompasses a variety of conditions that affect the heart. These conditions, as outlined in the Supplementary Table 1, range from conditions caused by lifestyle habits (hypertensive heart disease) to other conditions (conduction disorders). This results in our analyses including a larger set of cancer patients. Further, fatal heart disease as coded in SEER may be caused by the primary cancer itself; for example, a tumor may be pressing on the heart which leads to a heart attack. The cancer treatment may also be a cause of the heart issue that led to the patients' demise; for example, toxicity to the heart due to chemotherapy or radiation therapy may have caused a secondary heart condition that led to death[21]. Even though the death may be due to cancer, it will still be coded as death from diseases of the heart on the death certificate and thus in SEER. Further, there is no way to know which of these conditions that are included in the primary variable of interest was the ultimate cause of death.

There are some limitations to assess cause of death based on death certificates[22], outlined in the standardized reporting methods required for SEER, and the Supplementary Notes. It is predicted that the limitations of death certificate-based cause of death coding in SEER are similar to the limitations if cause of death were collected from another database. Of note, two studies have previously examined the validity and reliability of use of death certificates in survival and mortality-based studies using SEER[23,24]. In a study of patients with distant stage disease at diagnosis, reported cause of death matched the incident cancer diagnosis for 85% of patients[23]. The second study calculated observed-to-expected ratios (O/E) by cancer site as a measure of cause of death utility, where a favorable utility has an O/E close to 1.0; the calculated O/E by site were: breast = 0.97; colorectal = 0.98; lung = 0.90; melanoma = 1.07, which all suggest acceptable validity[24]. While the authors did not provide information regarding heart disease specifically, we assume death from heart disease would be as accurate. Using different methodology, both concluded that cause of death in large cancer registries like SEER is reliable with acceptable validity. Thus, we estimate that the cause of death is reliable in 85–98% of cases; it is on the lower end of this range in patients with multiple primaries or metastatic disease.

Another limitation is SEER does not have in-depth information regarding the type of cancer treatment the patient received beyond first treatment, including chemotherapy or radiation therapy; knowing complete treatment information may have allowed us to further investigate the cause and effect relationship between chemotherapy or radiation therapy and heart disease[13]. Additionally, we only included data from 1992 to 2015. Data in the SEER 18 Registries dates back to the 1970s. However, treatment paradigms for heart disease have changed drastically since this time period. Better treatment options have resulted in fewer patients dying from previously undetected heart conditions. Data was also collected starting in 1992 due to PSA testing and mammography being used more frequently, starting in the late 1980s/early 1990s[25]. This resulted in more patients being diagnosed with prostate and breast cancer, increasing the population included in our analyses and making the population more representative of the distribution of cancer patients today.

In conclusion, among 7,529,481 cancer patients, heart disease-specific mortality rate was 10.61 per 10,000-person years, and the SMR was 2.24 (95% CI, 2.23, 2.25). Patients with prostate, colorectum, breast, and lung cancer contribute to the plurality of patients dying of fatal heart disease. Odds of death from heart disease differs by age at diagnosis, sex, race, marital status, stage of primary cancer, year of diagnosis, and whether the patient received surgery. Among those diagnosed at <40 years of age, the plurality of heart disease deaths occurs in patients treated for breast cancer and lymphomas; if ≥40, from cancers of the prostate, breast, colorectum, and lung.

For almost all cancers survivors, the risk of fatal heart disease increases with time. These findings should be used to develop comprehensive guidelines regarding the prevention and care of heart disease in cancer survivors.

## Methods

**Data acquisition**. Patients with invasive cancer, diagnosed between 1992 and 2015, were abstracted from the National Cancer Institute's Surveillance, Epidemiology, and End Results (SEER) program[26,27]. The overview and limitations of the database and the methods are described in Supplementary Notes[28–31]. SEER collects data from 28% of the US population via a network of population-based incident tumor registries from geographically distinct regions in the US[26,27]. For the current analysis, the SEER 18 and 13 registries were used. The SEER registry includes data on age at diagnosis, sex, race, marital status, and year of diagnosis. SEER*Stat 8.3.5 was used for analysis[26]. All patients with an invasive cancer diagnosis were included. Patients diagnosed with cancer via autopsy or death certificate (<1.5% of patients) were excluded. Time in SEER is measured in months, with the minimum time to any event equal to 1 month. These data are freely available via the National Cancer Institute SEER program and are de-identified.

Mortality codes in SEER are assigned from death certificates, completed by the doctor caring for the patient at the time of demise. For the purposes of this study, patients were considered to have died of heart disease if the death certificate stated any of the International Classification of Diseases 9 (ICD-9) or ICD-10 codes included in the Supplementary Table 1. Further details of fatal heart disease are unknown because the specific ICD-9/-10 code for the condition that led to the patient's demise is unavailable; therefore, it is unknown if heart disease was due to a myocardial infarction, due to long-term congestive heart failure, or if heart disease was peri-operative, for example. SEER provides some subclassifications of "cardiovascular diseases," but it does not provide more detailed subclassifications of "heart disease," in part because (1) patients may die of multiple subclassifications of "heart disease" (e.g. myocardial infarction leading acute on chronic left sided heart failure), and (2) on death certificates in the United States, only one cause of death may be listed. Given this heterogeneity, in SEER, all of the subclassifications of "heart disease" are marked collectively as "heart disease." Notably, SEER does not code comorbidities or diagnoses associated with fatal heart disease, including smoking status, lipid levels, triglyceride levels, blood sugars, or prior events. The observed associations between cancer and fatal heart disease may be confounded by hypercholesterolemia, the use of medications, and patient lifestyle factors, but we are unable to control for these factors in the current work. These are limitations of the analysis and limit the interpretability of the results. While data was collected on all cancer sites available in SEER, the top 12 cancers based on highest overall person–years at risk are presented in the figures.

For objective 1, we calculated SMRs, which provide the relative risk of death for patients with cancer as compared to all US residents, stratified by cancer subgroup[26,32,33]. Although SEER data extends back to 1972, SMRs were calculated for the time period 1992–2015 because the identification and treatment of fatal heart disease has changed drastically over the decades, and the current time range would provide more contemporary outcomes while still providing information on long-term survival. Data were characterized with SMRs adjusted by age, race, and sex to the US population over the same time[31,34]. Five-year age categories were used for standardization using SEER*Stat 8.3.5 and Microsoft Excel 15.0.4 (Microsoft, Redmond, WA)[33,35,36]. SMRs should not be compared to each other, since they compare the risk of fatal heart disease in the group of interest vs. the risk in the standard population, and the standard population may be different among groups; further, SMRs and their confidence intervals depend on person years at risk, and if the incidence of a cancer is low and the survival is limited (e.g. liver cancer in the US), the CI may be wide—for these cancers, we do not report SMRs. The absolute and relative rate of fatal heart disease cases per patient age group (at time of diagnosis) and other patient characteristics were calculated.

For objective 2, a logistic regression model was used to obtain ORs with 95% CIs, calculated based on the number of observed events per patient subgroup, also for the time period 1992–2015. Further, we performed a survival analysis using a Cox proportional hazards model to calculate HRs, with the survival time being from diagnosis until fatal heart disease, and non-heart disease-related deaths plus living patients being censored. In the analyses for both objective 1 and objective 2 of this manuscript, patients who die of their cancer are no longer at risk for a fatal heart disease, and this is considered in the SMRs, ORs, and HRs.

**Reporting summary**. Further information on research design is available in the Nature Research Reporting Summary linked to this article.

## Data availability

The source data underlying Table 1, Figs. 1–3, and Supplementary Figs. 1 and 2, are provided as a Source Data file. Any additional information can be found within the supplementary information files or may be requested from the corresponding author. The data obtained for the current project from the SEER database are freely accessible to the public. We comply with all relevant ethical regulations. The datasets generated and analyzed during the current study are available in the SEER repository

(https://seer.cancer.gov/seerstat/). The study was exempt from IRB review as these data are freely available via the National Cancer Institute SEER program. There are no participants in the study, and thus there is no consent form.

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

## Author contributions

All authors had full access to all of the data in the study and take responsibility for the integrity of the data and the accuracy of the data analysis. Study concept and design: N.Z., K.C.S., Y.Z. Acquisition, analysis, and interpretation of data: N.Z., K.C.S., Y.Z. Drafting of the paper: N.Z., K.C.S. Critical revision of the paper for important intellectual content: K.C.S., Y.Z., K.S., L.S., D.T., V.C., N.Z. Statistical analysis: Y.Z., N.Z., K.C.S. Obtained funding: N/A. Administrative, technical, or material support: N.Z. Study supervision: N.Z.

## Competing interests

D.M.T reports clinical trial research support from Novocure, and publishing fees from Springer Inc. for projects outside the submitted work. K.S. reports funding from the National Center for Advancing Translational Sciences (NCATS) at the National Institute of Health [grant numbers 5 UL1 TR002014, 5 KL2 TR002015] for projects outside the submitted work. Novocure, Springer Inc., and the National Center for Advancing Translational Sciences had no role in the design of this study nor the execution, analyses, interpretation of the data, or decision to submit results. N.G.Z. is supported by the National Institutes of Health LRP 1 L30 CA231572-01 and the American Cancer Society, CSDG-CCE 133738. N.G.Z. received personal fees from Springer Nature, Inc for his textbook Absolute Clinical Radiation Oncology Review. He has also received payments from Weatherby Healthcare. Other authors declare no competing risks.
