## [Peer Review File · Nature Communications]

Reviewers' comments:

Reviewer #1 (Remarks to the Author):

Stolfus and colleagues conducted a retrospective, population-based cohort study on “fatal heart disease” of cancer patients captured in the SEER database. Cancer patients had a more than 2-fold higher cardiovascular mortality than non-cancer patients (SMR). The highest CV-SMR was seen in patients diagnosed with cancer at ages <39 (>40x higher SMR). On the contrary, the highest risk of CV mortality in cancer patients was seen in those >80 years. Among patients <40 years, breast cancer and lymphoma are the most "CV-fatal", among those >=40 years, it is breast, prostate, colorectal, and lung.

The term “fatal heart disease” is unique. In the methods section, the authors state that patients were considered to have died of heart disease if the death certificate stated any of the ICD-9/-10 code-related cardiovascular entities. Further details were unavailable and it was specified that it could not be discerned if heart disease was myocardial infarction, heart failure, or peri-operative. Do they mean if the cause of death was due to MI or HF or if the diagnosis of heart disease was related to it? These might be semantics, but heart disease versus cardiac mortality versus cardiovascular mortality are distinct and need to be accurately described herein.

Also, could any CVD on the death certificate still be considered and counted towards cause of death? Details on the SEER database and the data abstraction should be provided. The authors allude to this in the limitations on page 14.

The limitations of death certificates are well known. How accurate are they in the SEER database. Any prior publication that correlated SEER database cause of death versus verified cause of death would be helpful.

Fatal heart disease could also describe a heart condition that could ultimately be fatal, e.g. MI, HF. Do they authors mean and could cardiac mortality be a better term?

The umbrella term they use (line 86), is indeed, unusual terminology.

If cardiac mortality, do the authors have data on non-cardiac mortality? This might be a competing risk. SEER should be able to provide cancer-related risk of mortality and also other causes of mortality, e.g. after lymphoma, there are three phases on mortality, initial increase due to relapse,

subsequent plateau (when secondary cancers and complications progress), and finally increase in mortality due to “mature” secondary malignancies and CVD. Could this be reconstructed here? The authors show very nicely what they describe as a “U-shaped” dynamic.

Details on the source for the general population should be provided. Which data were obtained? Were they abstracted and coded the same way as they were in SEER? How can we be sure we can compare them? What is the congruence across the registries used?

Line 83, the reference is likely to the ESC position paper, which is not a practice guideline.

Line 89/90, could this be redundant information?

Line 93, cardiac mortality in cancer patients versus general population – as subtitle?

Line 106, why is the SMR higher in those diagnosed 2011-2015? Like age, there is a counter-trend in the HR among cancer patients (i.e. risk decreases from 1992 to 2015). There is room for speculation, but interesting to hear from the authors.

Line 121, cardiac mortality across cancer subgroups – as subtitle?

Line 122, this is the terminology alluded to above “died of fatal heart disease”, does this mean they died of MI, which can be fatal? Could they have died of mild TR, which is benign? The nature of the heart disease and the nature of cardiac mortality in this population should be better defined.

Line 123, do the data indicate that older cancer patients do not have a higher risk of cardiovascular mortality compared with the general population as CV mortality is increasing with age and among cancer patients only, still those <80 have the highest risk of CV mortality (>40 times higher risk than <39 years)? Who has the highest risk (>40x elevated)? It seems to be just a matter of the comparison group, no not? How drastic then is the issue in younger cancer patients? Is it overestimated? Line 136 seems to contradict the message in lines 99-103.

Line 171 and line 181, the objectives emerge later in the methods sections, the introduction does not highlight them as objective 1 and objective 2, but provides a more narrative approach. It might be advisable to rephrase.

Reviewer #3 (Remarks to the Author):

This is an interesting paper which has reported patterns of mortality from cause of fatal heart disease among cancer patients based on the SEER dataset. However I have some comments regarding the methods below.

1. Page 3. The statement "Our purpose was to identify cancer patients at highest risk of fatal heart disease compared

to the general population and other cancer patients.". Unclear about "cancer patients". You may clarify this.

2. Page 3. The statement "the rate of fatal heart disease was 10.61/10,000-person years". Unclear about "the rate". Incidence rate or mortality rate ? Please clarify this.

3. Line 320. "SMRs should not be compared .., since they compare the relative risk vs. the standard population, ..". Unclear about "compare the relative risk vs. the standard population".

4. Methods section and Table 2. It appears that the logistic regression was used to calculate ORs. However the logistic regression was not omitted in Methods section. Please provide the rationale for logistic regression methods given the nature of study design of the present study. Is it necessary to report both ORs and HRs?

5. Line 471. The statement "Figure 2. Age adjusted incidence rates per 100,000 for fatal heart disease by cancer subsite".

Regarding "Incidence rates", do you mean the mortality rates ?

6. Figures 2 and 3. They are not easy to follow. For example, Figure 2 is overcrowded. They should be improved to make your observed patterns more obvious. In addition, the observed time pattern based on Figure 3 may need further formal statistical test.

Reviewers' comments:

Reviewer #1 (Remarks to the Author):

1. Stolfus and colleagues conducted a retrospective, population-based cohort study on “fatal heart disease” of cancer patients captured in the SEER database. Cancer patients had a more than 2-fold higher cardiovascular mortality than non-cancer patients (SMR). The highest CV-SMR was seen in patients diagnosed with cancer at ages <39 (>40x higher SMR). On the contrary, the highest risk of CV mortality in cancer patients was seen in those >80 years. Among patients <40 years, breast cancer and lymphoma are the most "CV-fatal", among those >=40 years, it is breast, prostate, colorectal, and lung.

2. The term “fatal heart disease” is unique. In the methods section. the authors state that patients were considered to have died of heart disease if the death certificate stated any of the ICD-9/-10 code-related cardiovascular entities. Further details were unavailable and it was specified that it could not be discerned if heart disease was myocardial infarction, heart failure, or peri-operative. Do they mean if the cause of death was due to MI or HF or if the diagnosis of heart disease was related to it? These might be semantics, but heart disease versus cardiac mortality versus cardiovascular mortality are distinct and need to be accurately described herein.

AUTHOR RESPONSE: Thank you for your helpful comments. For patient level data, cause of death is coded in SEER in the following categories which, in combination, are defined as “cardiovascular disease”: diseases of the heart; hypertension without heart disease; cerebrovascular diseases; atherosclerosis; aortic aneurysm and dissection; other disease of arteries, arterioles, capillaries.

For the current work, we were most interested in “diseases of the heart,” which is most frequently called “heart disease” in the clinic. In SEER, “diseases of the heart” is listed as the cause of death if the death certificate reported one of the ICD-9/-10 codes reported in the supplementary information, which include the following codes, provided in the supplementary file:

Supplementary Table 1. ICD-9 and ICD-10 codes used for defining fatal heart disease.

ICD-9-CM diagnosis coding	Disease/condition	ICD-10-CM diagnosis coding	Disease/condition
390	Rheumatic fever without mention of heart involvement	I00	Acute rheumatic fever without heart involvement
391	Rheumatic fever with heart involvement	I01	Rheumatic fever with heart involvement
392	Rheumatic chorea	I02	Rheumatic chorea
393	Chronic rheumatic pericarditis	I05	Rheumatic mitral valve diseases
394	Diseases of mitral valve	I06	Rheumatic aortic valve diseases

395	Diseases of aortic valve	I07	Rheumatic tricuspid valve diseases
396	Diseases of mitral and aortic valves	I08	Multiple valve diseases
397	Diseases of other endocardial structures	I09	Other rheumatic heart diseases
398	Other rheumatic heart disease	I11	Hypertensive heart disease
402	Hypertensive heart disease	I13	Hypertensive heart and chronic kidney disease
404	Hypertensive heart and chronic kidney disease	I20	Angina pectoris
410	Acute myocardial infarction	I21	Acute myocardial infarction
411	Other acute and subacute forms of ischemic heart disease	I22	Subsequent ST elevation (STEMI) and non-ST elevation (NSTEMI) myocardial infarction
412	Old myocardial infarction	I23	Certain current complications following ST elevation (STEMI) and non-ST elevation (NSTEMI) myocardial infarction (within the 28 day period)
413	Angina pectoris	I24	Other acute ischemic heart diseases
414	Other forms of chronic ischemic heart disease	I25	Chronic ischemic heart disease
415	Acute pulmonary heart disease	I26	Pulmonary embolism
416	Chronic pulmonary heart disease	I27	Other pulmonary heart diseases
417	Other diseases of pulmonary circulation	I28	Other diseases of pulmonary vessels
420	Acute pericarditis	I30	Acute pericarditis
421	Acute and subacute endocarditis	I31	Other diseases of pericardium
422	Acute myocarditis	I32	Pericarditis in diseases classified elsewhere
423	Other diseases of pericardium	I33	Acute and subacute endocarditis
424	Other diseases of endocardium	I34	Nonrheumatic mitral valve disorders
425	Cardiomyopathy	I35	Nonrheumatic aortic valve disorders
426	Conduction disorders	I36	Nonrheumatic tricuspid valve disorders
427	Cardiac dysrhythmias	I37	Nonrheumatic pulmonary valve disorders
428	Heart failure	I38	Endocarditis, valve unspecified
429	Ill-defined descriptions and complications of heart disease	I39	Endocarditis and heart valve disorders in diseases classified elsewhere
		I40	Acute myocarditis
		I41	Myocarditis in diseases classified elsewhere
		I42	Cardiomyopathy
		I43	Cardiomyopathy in disease classified elsewhere
		I44	Atrioventricular and left bundle-branch block
		I45	Other conduction disorders
		I46	Cardiac arrest

		I47	Paroxysmal tachycardia
		I48	Atrial fibrillation and flutter
		I49	Other cardiac arrhythmias
		I50	Heart failure
		I51	Complications and ill-defined descriptions of heart disease

To clarify this intricacy of the SEER naming system we now state the following:

“Further details of fatal heart disease are unknown because the specific ICD-9/-10 code for the condition that led to the patient’s demise is unavailable; therefore, it is unknown if heart disease was due to a myocardial infarction, due to long-term congestive heart failure, or if heart disease was peri-operative, for example. SEER provides some subclassifications of “cardiovascular diseases,” but it does not provide more detailed subclassifications of “heart disease,” in part because (1) patients may die of multiple subclassifications of “heart disease” (e.g. myocardial infarction leading acute on chronic left sided heart failure), and (2) on death certificates in the United States, only one cause of death may be listed. Given this heterogeneity, in SEER, all of the subclassifications of “heart disease” are marked collectively as “heart disease.”

Please also note that our laboratory is interested in characterizing many of these competing risks of death, and we have a separate manuscript accepted at *Nature Communications*, “Fatal stroke among cancer patients,” which characterizes risk of stroke among those patients. The characteristics for stroke are very different than those of heart disease, so they are reported separately.

3. Also, could any CVD on the death certificate still be considered and counted towards cause of death? Details on the SEER database and the data abstraction should be provided. The authors allude to this in the limitations on page 14.

AUTHOR RESPONSE: Thank you for your helpful comments. In SEER, there are 6 death classifications related to cardiovascular disease (CVD). They include: diseases of the heart (focus of this study); hypertension without heart disease; cerebrovascular diseases (i.e. stroke); atherosclerosis; aortic aneurysm and dissection; other diseases of arteries, arterioles, capillaries. Therefore, not any CVD on the death certificate is considered and counted towards cause of death, simply those in the “diseases of the heart” category as specified in the ICD-9/-10 codes included in the supplementary information.

To clarify, we have added the following to the Supplementary Notes:

“Cardiovascular-related deaths can be classified as six different overarching categories according to SEER. They include: diseases of the heart; hypertension without heart disease; cerebrovascular diseases; atherosclerosis; aortic aneurysm and dissection; other diseases of arteries, arterioles, capillaries. The patients included in the present study are

coded as dying from diseases of the heart. A further breakdown of the ICD-9/-10 codes that fall into this category are provided in Supplementary Table 1.”

4. The limitations of death certificates are well known. How accurate are they in the SEER database. Any prior publication that correlated SEER database cause of death versus verified cause of death would be helpful.

AUTHOR RESPONSE: Thank you for your helpful comments. The reporting methods for SEER death certificates are very rigorous and controlled, as highlighted in the Supplementary Notes, provided below.

“Quality assurance and completeness

SEER undergoes quality assurance using systematic, standardized, and periodic data collection procedure for all defined members of a defined cohort is performed to avoid surveillance bias.¹ The case-finding audits are performed by a qualified member from each SEER registry under the direction of members of the National Cancer Institute. Auditors create an abstract that contains the primary site and the case finding source.² When performing audits, SEER adheres to two basic principles: auditing high quantity and high risk data. High quantity refers to disease sites that have the highest incidence and prevalence (e.g. breast, prostate, lung, colorectum); as well facilities that contribute the greatest percent of cases to the central database. Additionally, pathology laboratories are selected to review tissue from patients not seen at that hospital. High risk refers to cases that are likely to be miscoded (e.g. head and neck, hematopoietic diseases); compliance to new rules; and newly-reportable diseases.

Defining the cause of death

Mortality codes in SEER are assigned from death certificates, completed by the doctor caring for the patient at the time of demise. There is no single best method for calculating survival from cancer in the SEER program.³ Different methods can give different outcomes, but for most variants considered the differences are small. For heart disease, there may be some discrepancy in the cause of death, since the death may be because of the cancer itself, the cancer treatment, underlying heart disease, or a combination.”

For these reasons, it is predicted that the limitations of cause of death coding in SEER based on death certificate are similar to the limitations if they were pulled from another database.

To address your helpful comment in the manuscript, we have added the following to the discussion:

“There are some limitations to assess cause of death based on death certificates⁴, outlined in the standardized reporting methods required for SEER, and the Supplementary Notes. It is predicted that the limitations of death certificate-based cause of death coding in SEER are similar to the limitations if cause of death were collected from another database. Of note, two studies have previously examined the validity and reliability of use of death certificates in survival and mortality-based studies using SEER

^{5 6}. In a study of patients with distant stage disease at diagnosis, reported cause of death matched the incident cancer diagnosis for 85% of patients⁵. The second study calculated observed-to-expected ratios (O/E) by cancer site as a measure of cause of death utility, where a favorable utility has an O/E close to 1.0; the calculated O/E by site were: breast = 0.97; colorectal = 0.98; lung = 0.90; melanoma = 1.07, which all suggest acceptable validity⁶. While the authors did not provide information regarding heart disease specifically, we assume death from heart disease would be as accurate. Using different methodology, both concluded that cause of death in large cancer registries like SEER is reliable with acceptable validity. Thus, we estimate that the cause of death is reliable in 85-98% of cases; it is on the lower end of this range in patients with multiple primaries or metastatic disease.”

5. Fatal heart disease could also describe a heart condition that could ultimately be fatal, e.g. MI, HF. Do they authors mean and could cardiac mortality be a better term?
The umbrella term they use (line 86), is indeed, unusual terminology.

AUTHOR RESPONSE: Thank you for your helpful feedback. We appreciate your comments and recognize the complexities of using the umbrella term “fatal heart disease”. We chose this term for three reasons. First, it is the name used in the SEER database. Second, it is the terminology used in death certificates. Third, we discussed our decision to use this terminology with the chief of cardiology at Penn State Milton S. Hershey Medical Center, Dr. Lawrence Sinoway, and the manuscript co-authors and they were agreeable.

6. If cardiac mortality, do the authors have data on non-cardiac mortality? This might be a competing risk. SEER should be able to provide cancer-related risk of mortality and also other causes of mortality, e.g. after lymphoma, there are three phases on mortality, initial increase due to relapse, subsequent plateau (when secondary cancers and complications progress), and finally increase in mortality due to “mature” secondary malignancies and CVD. Could this be reconstructed here? The authors show very nicely what they describe as a “U-shaped” dynamic.

AUTHOR RESPONSE: Thank you for your helpful feedback. We have performed additional analyses, and we have created the following figures, added to the manuscript as supplementary figures 1 and 2, to address your comments. Each individual graph represents patients diagnosed with a different primary cancer, as indicated in the graph title. The y-axis represents the observed number of deaths and the x-axis represents the years since primary cancer diagnosis. The four colors represent the following causes of death: blue = primary cancer; orange = diseases of the heart; gray = non-index cancer; yellow = other cause of death. These other causes of death include any non-cancer related death that is recorded in the SEER database.

Supplementary Figure 1

Supplementary Figure 2

The manuscript now includes the following text:

“Additionally, **Supplementary Figures 1 and 2** display the observed number of deaths for each of the primary cancer sites presented, comparing the following cause of death groupings: primary cancer, diseases of the heart, non-index cancer, and other causes of death.”

7. Details on the source for the general population should be provided. Which data were obtained? Were they abstracted and coded the same way as they were in SEER? How can we be sure we can compare them? What is the congruence across the registries used?

AUTHOR RESPONSE: Thank you for your helpful feedback. US population census data are used as a comparison when calculating standardized mortality ratios (SMRs) by the SEER*Stat program. The exact characteristics of this population differ based on the years and/or age groups presented. When necessary, the 2000 US standard population from the census is used for age-adjusted statistics. County-level population data is available for download from the SEER website in a variety of formats (<https://seer.cancer.gov/popdata/download.html>). However, this was not necessary when calculating the SMRs for this project.

8. Line 83, the reference is likely to the ESC position paper, which is not a practice guideline.

AUTHOR RESPONSE: Thank you for your helpful feedback. The ESC does not currently have published guidelines for cardio-oncology care. To clarify this in our manuscript, we now state “International organizations, such as the European Society for Cardiology, recognize the need for such guidelines due to the complex nature of the relationship between cancer and cardiovascular health, and have published materials to assist clinicians in the care of this patient population until formal guidelines are available⁷”, with reference to the ESC position paper.

9. Line 89/90, could this be redundant information?

AUTHOR RESPONSE: Thank you for your helpful feedback. While possibly redundant, we feel that this paragraph is necessary for presenting the overarching goal and establishing the objectives of the study. We have changed the last sentence of the last paragraph in the introduction to now read “Our overarching goal is to identify subgroups of cancer patients at greatest risk of fatal heart disease.”

10. Line 93, cardiac mortality in cancer patients versus general population – as subtitle?

AUTHOR RESPONSE: Thank you for your helpful feedback. We have changed the subtitle to now read “Fatal heart disease in cancer patients vs general population”.

11. Line 106, why is the SMR higher in those diagnosed 2011-2015? Like age, there is a counter-trend in the HR among cancer patients (i.e. risk decreases from 1992 to 2015). There is room for speculation, but interesting to hear from the authors.

AUTHOR RESPONSE: Thank you for your helpful comments. The SMR calculations use the general population as the comparison group, while the HR calculations use other cancer patients as the comparison group. Thus, it is difficult to compare these different statistical analyses.

SMRs equal the observed/expected number of events. For the SMRs, in the general population, you are more likely to see an increased number of cases of fatal heart disease with more follow-up time (i.e. in those diagnosed from 1992-2000) because there is more opportunity to experience an event. Thus, the expected number of events is higher when calculating the SMR for this time period. The higher SMR in the later time period suggests that more cancer patients die from heart disease than the general population in this time period, potentially because the population of cancer patients are inherently sicker than the general population.

For the HR calculations, the reference group is other cancer patients who have experienced fatal heart disease. Thus, the two groups being compared (patients diagnosed from 1992-2000 vs. patients diagnosed from 2011-2015) are more similar than if one group were compared to the general population. In this situation, patients diagnosed in the earlier time period have more opportunity to experience the event of interest, thus making those more recently diagnosed appear to have a protective effect (HR = 0.63).

In addition, we have also added the following to our supplementary notes to provide further information regarding the intricacies of SMRs:

“Calculating Standardized Mortality Ratios

SMRs consist of two measures: (observed number of events, during time at risk) / (expected number of events in the reference population, during time at risk). SMRs may be calculated as a function of different times at risk, including time after diagnosis (i.e. the latency period) or age at diagnosis. When SMRs are calculated as a function of time after diagnosis, they provide the relative risk of death from one particular cause vs. the reference population. The reference population changes depending on the population and the time period. Thus, SMRs should not be compared to one another, and they would be expected to vary over different time periods or with different patient populations. Further, calculated SMRs may differ when using different SEER databases because (1) the observed number of events of interest among cancer patients may change, and (2) the number of events of interest in the reference population (i.e. the United States) also changes over the years.

Latency Exclusion Periods in Standardized Mortality Ratios

For SMRs calculated as a function of follow up time, SMRs during each window of time (e.g. at 1 year after diagnosis, 1-5 years after diagnosis, etc.) depend on the time at risk. With longer time at risk and more observed events, the confidence intervals become smaller, and measurements are more accurate. With a short time at risk (e.g. the first few months after diagnosis), or very few events (e.g. suicide), or among a niche patient cohort (e.g. Hodgkin lymphoma), the confidence intervals can widen dramatically.

In the first few months after diagnosis of cancer, patients often have an “introduction to the medical system;” i.e. a patient living in a rural area comes to a hospital where they are diagnosed with cancer, as well as many other comorbidities like heart disease, lung dysfunction, kidney failure, etc. The patient may die of any of these within a few months, but estimating the observed versus expected rate of death becomes difficult, and the confidence intervals for an SMR naturally widen. Thus, some researchers, including our team, sometimes elect to exclude the first 2 months from the SMR calculations. While SMRs may actually be very high during this time, the confidence intervals are so wide that an accurate measure is not meaningful. Moreover, the absolute number of observed events in this time may be rather low, especially when the event of interest is rare. Thus, the overall SMRs for the entire follow up period (with or without the latency periods) tend to be relatively similar.”

12. Line 121, cardiac mortality across cancer subgroups – as subtitle?

AUTHOR RESPONSE: Thank you for your helpful feedback. We have changed the subtitle to now read “Fatal heart disease across cancer subgroups”.

13. Line 122, this is the terminology alluded to above “died of fatal heart disease”, does this mean they died of MI, which can be fatal? Could they have died of mild TR, which is benign? The nature of the heart disease and the nature of cardiac mortality in this population should be better defined.

AUTHOR RESPONSE: Thank you for your helpful feedback. We have changed this phrase throughout the paper to now read “died of heart disease”, as died of fatal heart disease is redundant. Also, this change also refers to our previously defined umbrella term/SEER terminology, meaning that death could be a result of any of the provided ICD-9/-10 codes. This has been explained further in response to comments #2 and #3.

14. Line 123, do the data indicate that older cancer patients do not have a higher risk of cardiovascular mortality compared with the general population as CV mortality is increasing with age and among cancer patients only, still those <80 have the highest risk of CV mortality (>40 times higher risk than <39 years)? Who has the highest risk (>40x elevated)? It seems to be just a matter of the comparison group, no not? How drastic then is the issue in younger cancer patients? Is it overestimated? Line 136 seems to contradict the message in lines 99-103.

AUTHOR RESPONSE: Thank you for your helpful feedback. Similar to the explanation regarding differences by year of diagnosis, it is expected to see different results when comparing SMRs vs ORs/HRs. As stated, the findings from the SMRs suggest that older cancer patients do not have much higher risk of fatal heart disease compared with the general population (still higher with an SMR = 1.73). This may be because risk of fatal heart disease increases with age regardless of whether a patient has cancer or not. These events of fatal heart disease are seen more in the cancer population than in the general population in the younger age groups (SMR = 43.8 for ≤39 years old), suggesting there is something about their cancer that is increasing their susceptibility to death from heart disease.

However, when looking at the ORs and HRs, the comparison group is other cancer patients. Thus, among all cancer patients who experience fatal heart disease, the older the patient is at diagnosis, the more likely they are to die of heart disease. Again, this can be explained in part that risk of heart disease increases with age. It is expected that trends in these results are opposite those of the SMRs because the comparison group is different.

In summary, when compared to the general population, younger patients have an elevated risk of fatal heart disease. When compared to other cancer patients, older patients have an elevated risk of fatal heart disease. The risk in these scenarios varies because of the reference group. For more details, please see our response to comment #11.

Finally, we have updated our analysis to include two other pediatric tumor populations that have a relatively low cancer-specific mortality and high mortality from heart disease, Hodgkin lymphoma and testicular cancer. The updated results are shown in Figures 1-3.

15. Line 171 and line 181, the objectives emerge later in the methods sections, the introduction does not highlight them as objective 1 and objective 2, but provides a more narrative approach. It might be advisable to rephrase.

AUTHOR RESPONSE: Thank you for your helpful feedback. As the manuscript currently stands, lines 87-89 of the introduction state the numbered objectives. For clarity, we have revised this line to state “Our objectives are to identify cancer patients at highest risk of fatal heart disease compared to (1) the general population, using standardized mortality ratios (SMRs), and (2) other cancer patients at risk of death during the study time period, using odds ratios (ORs) and hazard ratios (HRs).”

Reviewer #3 (Remarks to the Author):

This is an interesting paper which has reported patterns of mortality from cause of fatal heart disease among cancer patients based on the SEER dataset. However I have some comments regarding the methods below.

1. Page 3. The statement "Our purpose was to identify cancer patients at highest risk of fatal heart disease compared to the general population and other cancer patients.". Unclear about "cancer patients". You may clarify this.

AUTHOR RESPONSE: Thank you for your helpful comments. We have changed this line to now read “Our purpose was to identify cancer patients at highest risk of fatal heart disease compared to (1) the general population and (2) other cancer patients at risk of death during the study time period.” This was also clarified in the second to last sentence in the last paragraph of the introduction, which now reads: “Our objectives are to identify cancer patients at highest risk of fatal heart disease compared to (1) the general population, using standardized mortality ratios (SMRs), and (2) other cancer patients at risk of death during the study time period, using odds ratios (ORs) and hazard ratios (HRs).”

The reason we compare to other cancer patients is because clinicians want to be able to identify the cancer patients (versus other cancer patients) who will be at highest risk of heart disease so they can be referred to cardiology if necessary. With the expanding world of cardio-oncology, we believe this work can help treatment facilities establish direct referral centers for patients at highest risk to die of heart disease.

2. Page 3. The statement "the rate of fatal heart disease was 10.61/10,000-person years". Unclear

about "the rate". Incidence rate or mortality rate ? Please clarify this.

AUTHOR RESPONSE: Thank you for your helpful feedback. We have changed this throughout for clarification. It now reads “the heart disease-specific mortality rate per 10,000-person years was 10.61”.

3. Line 320. "SMRs should not be compared .., since they compare the relative risk vs. the standard population, ..". Unclear about "compare the relative risk vs. the standard population".

AUTHOR RESPONSE: Thank you for your helpful feedback. We have changed the sentence to now read “SMRs should not be compared to each other, since they compare the risk of fatal heart disease in the group of interest vs. the risk in the standard population, and the standard population may be different among groups”. For more details, please see our response to reviewer 1, comment #11, which includes changes to the Supplementary Notes that expand upon the intricacies of SMRs.

4. Methods section and Table 2. It appears that the logistic regression was used to calculate ORs. However the logistic regression was not omitted in Methods section. Please provide the rationale for logistic regression methods given the nature of study design of the present study. Is it necessary to report both ORs and HRs?

AUTHOR RESPONSE: Thank you for your helpful feedback. The methods regarding objective 2 now state “For objective 2, a logistic regression model was used to obtain ORs with 95% CIs, calculated based on the number of observed events per patient subgroup, also for the time period 1992-2015.” We included both ORs and HRs because these tests allow us to account for the cause of death, and for survival time from cancer diagnosis until fatal heart disease, respectively.

The odds of a patient with localized disease dying from heart disease is greater than that of a patient with distant disease because the patient with distant disease is probably going to die from their cancer, thus they are not at risk of dying from heart disease. In comparison, the hazard ratio, which takes into account time to event, is less for patients with localized disease when compared to patients with distant disease. This can be explained by the idea that patients with localized disease are living longer than patients with distant disease, regardless of the cause of death. Having both the OR and HR allow two different, yet equally relevant interpretations of the results. Thus, it is not surprising that when comparing the ORs and HRs for stage by stage comparisons, the OR and HR will provide different values.

5. Line 471. The statement "Figure 2. Age adjusted incidence rates per 100,000 for fatal heart disease by cancer subsite".

Regarding "Incidence rates", do you mean the mortality rates ?

AUTHOR RESPONSE: Thank you for your helpful feedback. We have changed incidence rate to mortality rate where applicable throughout the manuscript, tables, and figures.

6. Figures 2 and 3. They are not easy to follow. For example, Figure 2 is overcrowded. They

should be improved to make your observed patterns more obvious. In addition, the observed time pattern based on Figure 3 may need further formal statistical test.

AUTHOR RESPONSE: Thank you for your helpful feedback. We have conducted further statistical tests of the data from Figure 3 using a trend test for change in proportion of deaths from primary cancer versus heart disease. We have provided this information in Supplementary Table 2, which can be found in the Supplementary Information. Additionally, we have added appropriate references to these analyses throughout the manuscript and figure legends, including the following in the results section: “Trend tests for changes in proportion of deaths from primary cancer versus heart disease are statistically significant for each site tested (chi-squared value range: 30.9-123,840; $P < 0.001$), as presented in **Supplementary Table 2.**”

Further, we have done analyses to develop cumulative incidence plots for deaths over time. For more information on these, please see our response to review 1, comment #6.

Supplementary Table 2. Trend test for change in proportion of deaths from primary cancer vs. heart disease.

Cancer Site	Chi-squared Value¹	P-value
All Sites	123840	<0.001
Prostate	4383.3	<0.001
Breast	7136.6	<0.001
Colon and Rectum	22884	<0.001
Bladder	5943.9	<0.001
Melanoma	2203.6	<0.001
Lung	12373	<0.001
Kidney	3514.1	<0.001
Endometrial	4167	<0.001
Leukemia	1717.8	<0.001
Oral Cavity And Pharynx	2744.1	<0.001
Myeloma	30.9	<0.001
Non-Hodgkin Lymphoma	4299.9	<0.001

1- DF = 1

We would also like to clarify the work of our lab and our currently submitted works. Our lab focuses in part on competing risks of death in cancer patients. We have submitted a manuscript to the *European Heart Journal* (recently accepted) and two manuscripts to *Nature Communications* (including the current manuscript on fatal heart disease [NCOMMS-19-26028], and another manuscript on stroke [NCOMMS-19-05266A], currently undergoing revisions) that builds on this theme. Below, we clarify how these manuscripts are completely separate analyses from one another, with different purposes and findings, and without overlap in data. If you have any questions about the works, please do not hesitate to contact us.

Manuscript title	Fatal heart disease in cancer patients	Stroke in cancer patients	Cardiovascular disease in cancer patients
Journal considering manuscript	Nature Communications NCOMMS-19-26028	Nature Communications NCOMMS-19-05266A	European Heart Journal EURHEARTJ-D-18-02987R2
Objectives	To characterize death from heart disease vs (1) general population and (2) other cancer patients	To characterize death from stroke vs (1) the general population and (2) other cancer patients	To characterize cardiovascular diseases in cancer patients as a function of (1) calendar year, (2) age at diagnosis, and (3) follow-up time after cancer diagnosis
Diseases included	Heart disease. Individual risk factors for death from disease are considered.	Stroke. Individual risk factors for death from disease are considered.	All cardiovascular diseases, including diseases of the heart, hypertension without heart disease, cerebrovascular diseases, aortic aneurysm and dissection, atherosclerosis, and other diseases of arteries, arterioles, capillaries. Diseases are generally considered as a whole, and individual risk factors for diseases are not considered.
Synopsis	In the contemporary era of cancer medicine (1990s - today), it is known that heart disease is a leading cause of death in cancer patients. However, it is unknown which cancer patients are at highest risk vs the general population or compared to other cancer patients. We find that cancer patients are at a persistent increased risk of fatal heart disease after diagnosis: for children, adolescents, and adults < 40 years old, the plurality of heart disease deaths is seen in breast cancer, testicular cancer, and lymphoma patients. In contrast, among adults ≥ 40 years old the plurality of heart disease deaths occurs in patients with cancer of the prostate, colorectum, breast, and lung. Compared to other	In the contemporary era of cancer medicine (1990s - today), it is known that certain cancer patients are at an increased risk of death from stroke. However, it is unknown which cancer patients are at highest risk vs the general population or compared to other cancer patients. We find that cancer patients are at a persistent increased risk of fatal stroke after diagnosis: for children, adolescents, and young adults (i.e. < 40 years old), the plurality of strokes is seen in brain tumor and lymphoma patients. In contrast, among older adults (i.e. >40 years old) the plurality of strokes occurs in patients with cancer of the prostate, breast, and	In the 1970s, a cancer diagnosis was generally considered a death sentence. Throughout the 1980s and 1990s, the cancer patient pool has included cancers with a lower risk of cancer specific mortality, and treatments for cancer have improved cure rates. Thus, many cancer survivors now die of competing causes, including cardiovascular diseases (heart disease, aneurysm, stroke, hypertension, etc). In this paper, we characterize the historical trends of deaths from cancer vs cardiovascular diseases. We find that in several sites, the risk of cardiovascular deaths now supersedes risk of death from index cancer (objective 1). We find that the relative risk (compared to the general population) of death from

	cancer patients, patients who are older, male, African American, and unmarried are at a greatest risk of fatal heart disease. For almost all cancers survivors, the risk of fatal heart disease follows a u-shaped phenomenon, highest in the first year after diagnosis, then decreasing, then increasing with time.	colorectum. Compared to other cancer patients, patients who are older, male, African American, and unmarried are at a greatest risk of fatal stroke. Risk of stroke tends to increase with younger age and longer follow up time. Young patients with lymphomas and brain tumors are at a particularly high risk with longer follow up time.	cardiovascular disease is highest when patients are diagnosed at a young age (objective 2). The first year after diagnosis represents a period of high risk for death, and the cumulative incidence of cardiovascular disease increases with follow up time differently among cancers (objective 3). The factors placing patients at increased risk of death from specific cardiovascular diseases vs other cancer patients and vs the general population will be reported in future works. The work from this paper inspired my lab to focus on specific diseases, for future works.
Methods used per objective	(1) Standardized mortality ratios vs follow-up time (2) Individual patient level data used to calculate odds ratios, hazard ratios, and cumulative incidence plots (as requested by the reviewers)	(1) Standardized mortality ratios vs follow-up time, with minor focus on age at diagnosis (described below) (2) Individual patient level data used to calculate odds ratios and hazard ratios	(1) Incidence rates of death over time (i.e. decades of cancer diagnosis) (2) Standardized mortality ratios vs age at diagnosis (3) Standardized mortality ratios vs follow-up time
Methods not used	Incidence rate sessions over decades, standardized mortality ratios vs age at diagnosis	Incidence rate sessions over decades, cumulative incidence functions	Individual patient level data, odds ratios, hazard ratios
Country	United States	United States	United States
Years included	1992-2015	1992-2015	1973-2012, with subgroup analysis of 2000-2015
Principal databases used	Objective 1: Surveillance, Epidemiology, and End Results (SEER) - Standardized incidence rate session, focused on follow up time after diagnosis, cancer subsite, age Objective 2: Surveillance, Epidemiology, and End Results (SEER) - Case listing session	Objective 1: Surveillance, Epidemiology, and End Results (SEER) - Standardized incidence rate session, focused on follow up time after diagnosis, cancer subsite, age Objective 2: Surveillance, Epidemiology, and End Results (SEER) - Case listing session	Objective 1: Surveillance, Epidemiology, and End Results (SEER) - Incidence rate session, focused on year at death Objective 2: Surveillance, Epidemiology, and End Results (SEER) - Standardized incidence ratio session, focused on age at diagnosis Objective 3: Surveillance, Epidemiology, and End Results (SEER) – Standardized

			incidence ratio in SEER
Assessment of disease deaths over decades	No Methods used: N/A	No Methods used: N/A	Yes (Figure 1, Figure 2, Figure 3, Figure 4, e Figure 1, Table 1) Methods used: Incidence rate session in SEER
Assessment of cancers where death from disease of interest supersedes, is equal to, or is less than the number of deaths from cancer, among all survivors in the US	No Methods used: N/A	No Methods used: N/A	Yes (Figure 2-4)
Assessment of risk vs general population at age of diagnosis	No Methods used: N/A	Minor part of analysis (Figure 2), to clarify risk for pediatric cancer patients. Standardized incidence ratio in SEER.	Yes (Figure 5 A/B) Methods used: Standardized incidence ratio in SEER. Subgroup analysis of contemporary risk for 2000-2015.
Cumulative incidence function assessment of being alive vs death from all cardiovascular diseases vs death from other causes vs death from other cancer	Yes (Supplementary Figures 1-2, as requested by reviewers) Within cancer patients of each cancer Methods used: Cumulative incidence plots	No Methods used: N/A	Yes (eFigure 2) Within cancer patients of each cancer Methods: cumulative incidence function
Assessment of risk vs general population at time after diagnosis	Yes, with extensive focus on cancer disease subsites, time after diagnosis, with standardized mortality ratios with confidence intervals (Figure 1) Methods used: Standardized incidence ratio in SEER	Yes, with extensive focus on cancer disease subsites, time after diagnosis, with standardized mortality ratios with confidence intervals (Figure 1) Methods used: Standardized incidence ratio in SEER	Yes, general overview (Figure 5 C/D, Figure 6) Methods used: Standardized incidence ratio in SEER
(from previous prompt) pediatric tumors included	Yes, lymphomas, leukemias, brain tumors, testicular cancer	Yes, lymphomas, leukemias, brain tumors	No
Odds ratios and hazard ratios of death from disease of interest vs other cancer patients	Yes (Table 2) Methods used: SEER case listing session data used to generate individual patient level data to calculate odds ratios and hazard ratios	Yes (Table 2) Methods used: SEER case listing session data used to generate individual patient level data to calculate odds ratios and hazard ratios	No

		ratios	
Patient level data with consideration of sex, surgery, marital status, stage, race	Yes (Table 2). Included in calculation of odds ratios and hazard ratios	Yes (Table 2). Included in calculation of odds ratios and hazard ratios	No
Assessment of proportion of patients per age group and cancer site dead of disease	Yes (Figure 2) Methods used: SEER incidence rate session, focused on follow up time after diagnosis, cancer subsite, age	Yes (Figure 3) Methods used: SEER incidence rate session, focused on follow up time after diagnosis, cancer subsite, age	No
Assessment of observed deaths from disease of interest vs death from other cause	Yes (Figure 3) Methods: Observed number of deaths plotted as a function of follow up time	No	No

Thank you again for the chance to revise our manuscript. We believe that we have successfully addressed all concerns of our reviewers. If you request other edits or revisions, please do not hesitate to contact us.

Sincerely,

Nicholas G Zaorsky MD

Assistant Professor (Tenure-track), Department of Radiation Oncology, Penn State Cancer Institute, Hershey, PA

Assistant Professor, Department of Public Health Sciences, Penn State Health Milton S. Hershey Medical Center, Hershey, PA

Physician-leader, Radiation Oncology Genitourinary Cancer Program

Physician-leader, Radiation Oncology Research Program

500 University Drive

Hershey, PA 17033

USA. Tel: +1-717-531-8024

Fax: +1-717-531-0446

E-mail: nicholaszaorsky@gmail.com; nzaorsky@pennstatehealth.psu.edu

References

- 1 Park, H. S., Lloyd, S., Decker, R. H., Wilson, L. D. & Yu, J. B. Overview of the Surveillance, Epidemiology, and End Results database: evolution, data variables, and quality assurance. *Current problems in cancer* **36**, 183-190, doi:10.1016/j.currproblcancer.2012.03.007 (2012).
- 2 National Cancer Institute. *Casefinding Studies - SEER Quality Improvement.*, <<http://seer.cancer.gov/qi/tools/casefinding.html>> (2016).
- 3 Boer, R. *et al.* (Statistical Research and Applications Branch, NCI, Bethesda, MD).
- 4 Brooks, E. G. & Reed, K. D. Principles and Pitfalls: a Guide to Death Certification. *Clinical medicine & research* **13**, 74-82; quiz 83-74, doi:10.3121/cmr.2015.1276 (2015).
- 5 Lund, J. L., Harlan, L. C., Yabroff, K. R. & Warren, J. L. Should cause of death from the death certificate be used to examine cancer-specific survival? A study of patients with distant stage disease. *Cancer Invest* **28**, 758-764, doi:10.3109/07357901003630959 (2010).
- 6 Hu, C. Y., Xing, Y., Cormier, J. N. & Chang, G. J. Assessing the utility of cancer-registry-processed cause of death in calculating cancer-specific survival. *Cancer* **119**, 1900-1907, doi:10.1002/cncr.27968 (2013).
- 7 Zamorano, J. L. *et al.* 2016 ESC Position Paper on cancer treatments and cardiovascular toxicity developed under the auspices of the ESC Committee for Practice Guidelines: The Task Force for cancer treatments and cardiovascular toxicity of the European Society of Cardiology (ESC). *European heart journal* **37**, 2768-2801, doi:10.1093/eurheartj/ehw211 (2016).

REVIEWERS' COMMENTS:

Reviewer #1 (Remarks to the Author):

No additional comments

Reviewer #3 (Remarks to the Author):

I have no further comments.